# PKAc is not required for the preerythrocytic stages of *Plasmodium berghei*

Hadi Hasan Choudhary[1], Roshni Gupta[1], Satish Mishra[1,2] (ORCID)

*Plasmodium* sporozoites invade hepatocytes to initiate infection in the mammalian host. In the infected hepatocytes, sporozoites undergo rapid expansion and differentiation, resulting in the formation and release of thousands of invasive merozoites into the bloodstream. Both sporozoites and merozoites invade their host cells by activation of a signaling cascade followed by discharge of micronemal content. cAMP-dependent protein kinase catalytic subunit (PKAc)–mediated signaling plays an important role in merozoite invasion of erythrocytes, but its role during other stages of the parasite remains unknown. Becaused of the essentiality of PKAc in blood stages, we generated conditional mutants of *PKAc* by disrupting the gene in *Plasmodium berghei* sporozoites. The mutant salivary gland sporozoites were able to glide, invaded hepatocytes, and matured into hepatic merozoites which were released successfully from merosome, however failed to initiate blood stage infection when inoculated into mice. Our results demonstrate that malaria parasite complete preerythrocytic stages development without PKAc, raising the possibility that the PKAc independent signaling operates in preerythrocytic stages of *P. berghei*.

## Introduction

The malaria parasite *Plasmodium* is a protozoan pathogen belonging to the phylum Apicomplexa. Malaria infection is initiated by the bite of infected mosquitoes that inoculate sporozoites during a blood meal. Sporozoites rapidly migrate to the liver and invade hepatocytes where they replicate and develop into merozoites that invade and multiply within host red blood cells. Invasion of host cells by *Plasmodium* parasites is a complex multistep process mediated by intracellular signaling cascades and regulated secretion by micronemes and rhoptries (Cowman & Crabb, 2006). When parasites contact the host cell, timely secretion and recruitment of ligands to the parasite surface is critical for successful invasion (Baum, 2013). Signaling events regulate protein secretion from specialized organelles (Ejigiri & Sinnis, 2009), which is mediated by protein kinases. There are about 93 kinase-encoding genes identified in *Plasmodium falciparum* (Talevich et al, 2011), and they are classified into different families and groups (Hanks & Hunter, 1995; Loomis et al, 1997). Gene deletion studies have shown the involvement of different kinases in multiple stages of parasite development. These stages include sporozoite infectivity, blood stage schizogony, gametogenesis, ookinete migration, and maturation and oocyst formation (Doerig et al, 2008).

The malaria parasite repeatedly performs invasion and egress functions to infect new cells and sustain its life. cAMP/PKA signaling, known to be a major player in responding to host factors, is stimulated when parasite G-protein-coupled receptors receive extracellular signals and activate adenyl cyclases to produce 3′-5′ cAMP. In *Plasmodium* merozoites, cAMP/PKA signaling plays an essential role in the invasion of erythrocytes by regulating cytosolic $Ca^{++}$ levels and micronemal protein secretions (Dawn et al, 2014). Similarly, the PKAc homolog in *Toxoplasma gondii* (PKAc1) plays a role in invasion via regulation of intracellular calcium (Uboldi et al, 2018). The cAMP-dependent PKA is a key mediator of the signal transduction pathway and plays diverse roles in the cell. The cAMP is generated by adenylyl cyclase, which primarily activates PKA (Taylor et al, 1990). PKA is known to regulate different processes in eukaryotic cells. In a related apicomplexan parasite *T. gondii*, PKA has been shown to be involved in tachyzoite to bradyzoite development (Kirkman et al, 2001; Eaton et al, 2006; Hartmann et al, 2013). PKA phosphorylates apical membrane antigen 1 (AMA1) protein of *Plasmodium*, and this phosphorylation event is essential for the invasion of new erythrocytes by merozoites (Leykauf et al, 2010).

During the journey from the skin to the liver, sporozoites come in direct contact with blood and the cytosol of the host cell that likely influences their infectivity patterns. First, sporozoites get exposed to a low K$^+$ (5 mM) environment in blood when they enter capillary to reach the liver. After reaching the liver, sporozoites traverse through several cells where they are exposed to high K$^+$ (140 mM) concentrations in the cell cytosol. What host factors activate parasites are elusive; as merozoites encounter a low K$^+$ environment in blood, adenylyl cyclase β (ACβ) is activated, leading to rise in cAMP levels and activation of PKA, which regulates micronemal secretions (Dawn et al, 2014), whereas in sporozoites, uracil derivatives stimulate exocytosis in the presence of high K$^+$. The enzyme primarily responsible for exocytosis in sporozoites is adenylyl

---

[1]Division of Parasitology, CSIR-Central Drug Research Institute, Lucknow, India   [2]Academy of Scientific and Innovative Research, Ghaziabad, India

Correspondence: satish.mishra@cdri.res.in

cyclase *α* (AC*α*), which is a functional enzyme having a transmembrane K$^+$ channel and an intracellular adenylyl cyclase domain. AC*α* knockout sporozoites were impaired in exocytosis and showed 50% less hepatocyte infectivity in vivo. When sporozoites were exposed to different K$^+$ channel inhibitors, exocytosis was inhibited, suggesting that K$^+$ is required for the activation of exocytosis (Ono et al, 2008). Another study also reported that preincubation of sporozoites in a high K$^+$ environment increases the hepatocyte infectivity (Kumar et al, 2007). Another secondary messenger cGMP mediated by the parasite's cGMP-dependent protein kinase (PKG), which is a signaling hub and indispensable in blood stage, is required for sporozoite invasion and parasite egress from hepatocytes (Falae et al, 2010; Govindasamy et al, 2016).

In merozoites, Ca$^{++}$-mediated signaling pathways regulate microneme secretion (Dawn et al, 2014) and the role of the Ca$^{++}$ ionophore is also implicated in inducing apical regulated exocytosis in *Plasmodium* sporozoites (Mota et al, 2002), suggesting that Ca$^{++}$ signaling plays a central role during the invasion of both merozoites and sporozoites. In sporozoites, Ca$^{++}$ signaling is essential for exocytosis as preincubation of sporozoites with the Ca$^{++}$ chelator strongly inhibited exocytosis (Ono et al, 2008). Ca$^{++}$ signaling is mediated by the parasite's calcium-dependent protein kinases (CDPKs). Sporozoites carrying a deletion of the CDPK4 gene were defective in invasion of hepatocytes. Another calcium-dependent protein kinase CDPK1 suggested to be essential for the erythrocytic stage of *P. falciparum* (Tewari et al, 2010) is found to be dispensable throughout the life cycle of *Plasmodium berghei* (Jebiwott et al, 2013). Protein kinases and CDPK signaling are interconnected, and CDPK1 phosphorylates the PKA regulatory subunit and regulates PKA activity in the parasite (Kumar et al, 2017). Regulation of CDPK1 is controlled by phosphorylation by PKG (Alam et al, 2015). For the activation of the protease cascade, which is critical for parasite egress, CDPK5 cooperates with the PKG in *P. falciparum* (Bansal et al, 2016).

It is clear that PKAc and PKG signaling is required in *Plasmodium* blood stages (Taylor et al, 2010; Brochet et al, 2014; Dawn et al, 2014; Bushell et al, 2017). In addition, the role of PKG has already been established in preerythrocytic stages (Falae et al, 2010). Whether PKAc is also required during preerythrocytic stages is not understood. To understand this, we used a reverse genetic approach, and owing to the essentiality of *PKAc* in *Plasmodium* blood stages, conditional mutants were generated by using a yeast Flp/FRT-based conditional mutagenesis system. The *PKAc* gene was successfully deleted in midgut sporozoites, and we show that *PKAc* cKO sporozoites invaded hepatocytes, completed liver stage (LS) development, and formed merosomes. However, *PKAc* cKO merosomes failed to initiate blood stage infection, whereas parasites that escaped excision were able to infect erythrocytes. Here, we show that PKAc is indispensable during erythrocytic stages but not required in preerythrocytic stages.

# Results

### *PKAc* transcripts are up-regulated in merozoites and sporozoites

To determine the transcript level of *PKAc* in different stages of the parasites, quantitative real-time PCR was performed. We analyzed *PKAc* expression in mixed blood stages, schizonts, midgut sporozoites,

salivary gland (SG) sporozoites, and LSs at 24, 38, and 65 h. *PKAc* predominantly transcribes in hepatic and blood merozoites, although significant transcript levels were also present in sporozoites (Fig 1A). The high level of *PKAc* transcripts in merozoites concurs their reported role in merozoite invasion (Dawn et al, 2014). The presence of *PKAc* transcripts in SG sporozoites suggests their possible role in sporozoite invasion as reported earlier (Ono et al, 2008).

### Generation of *PKAc* conditional KO parasites

Because of the essentiality of *PKAc* in *Plasmodium* blood stages, an Flp/FRT-based conditional mutagenesis system of yeast was used to generate *PKAc* conditional KO (*PKAc* cKO) parasites (Choudhary et al, 2018; Combe et al, 2009; Falae et al, 2010; Lacroix et al, 2011). *PKAc* contains 5 exons and 4 introns (Fig 1B (a)), and the Flp recognition target (FRT) site was engineered in its first intron (Fig 1B (b)). For the construction of the targeting plasmid p*PKAc*/FRT, three fragments F1, F2, and F3 were cloned in a p3'TRAP-hDHFR-flirte plasmid in such a way that after recombination, the FRT site is incorporated in the intron and the TRAP 3'UTR drives the expression of the gene (Fig 1B (c)). The targeting fragment was transfected into the *P. berghei* TRAP/FlpL parent line, which expresses FlpL recombinase under the control of the TRAP promoter in mosquito stages and excises the DNA sequence flanked by FRT sites (Fig 1B (d)) ([Lacroix et al, 2011]; S Mishra, KA Kumar and P Sinnis, unpublished data). The integration of the targeting cassette at the correct locus was verified by diagnostic PCR using primers 1072/1216 and 1215/1073 for 5' and-3' integration, respectively, and the full cassette was amplified using primers 1409/1073, which differentiated WT and the recombinant locus by size (Fig 1C), Incorporation of the FRT site was also confirmed by amplifying a fragment using primers 1409/1074 and sequencing. Two independent clones of *PKAc* cKO parasites were selected after limiting dilutions and named clone c1 and c2. To analyze if swapping of 3'UTR of *PKAc* with TRAP had any effect on the kinetics of propagation between wild-type (TRAP/FlpL) and *PKAc* cKO, blood stage growth assay was performed, which revealed both groups propagated at similar rates (Fig 1D).

### *PKAc* deletion does not affect mosquito stage development

Next, we investigated if *PKAc* deletion affected parasite development in mosquito stages. For parasite transmission, mosquitoes were allowed to feed on mice infected with *PKAc* cKO or TRAP/FlpL parasites. Examination of infected midguts on day 14 post blood meal demonstrated that *PKAc* cKO parasites formed oocysts and oocyst-derived sporozoites, which were comparable with TRAP/FlpL (Fig 2A–C). We enumerated the SG sporozoites on day 18 post blood meal and observed normal number in both groups (Fig 2E). These results demonstrate that the deletion of *PKAc* does not affect sporogony and migration of sporozoite from oocysts to SGs.

### *PKAc*-deficient sporozoites are unable to initiate blood stage infection

To investigate infectivity of *PKAc* cKO sporozoites, C57BL/6 mice were injected with *PKAc* cKO and TRAP/FlpL SG sporozoites intravenously and appearance of the parasite in blood was

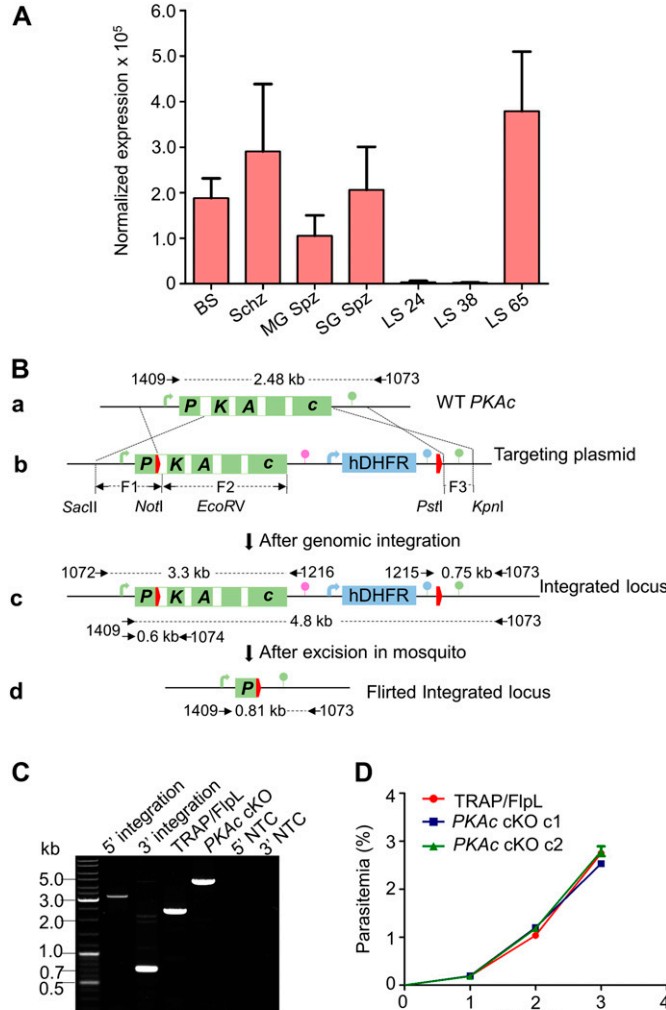

**Figure 1. *PKAc* expression analysis and generation of conditional knockout parasites.**
**(A)** The gene expression of *PKAc* was analyzed by quantitative PCR that revealed high level of expression in the blood (Schz) and hepatic (LS 65) schizonts. The expression was normalized with *P. berghei Hsp70*. BS; blood stages, Schz; schizonts, MG Spz, midgut sporozoites, SG Spz; SG sporozoites. Data are shown as means ± SEM (n = 3). **(B)** a. Schematic representation of the *PKAc* wild-type locus. b. The targeting plasmid p*PKAc*/FRT contains 1.2-kb F1, including 5′UTR of *PKAc*, first exon, and part of the first intron of *PKAc*, followed by the first FRT site (red thick arrow); F2 representing 1.6-kb remaining part of the gene, TRAP 3′UTR (pink lollipop), human dihydrofolate reductase (hDHFR) cassette, and second FRT site (red thick arrow); and 0.53-kb F3 showing 3′UTR of the *PKAc* gene. 5′ and 3′ regulatory sequences are represented by an arrow and lollipop, respectively. PKAc 3′UTR (green lollipop) and *P. berghei* DHFR/TS 3′UTR (blue lollipop). c. Following successful double crossover homologous recombination (as indicated by the dotted lines), the wild-type locus was replaced by the targeting vector sequence. d. Excision of the flirted locus after passing through mosquito. **(C)** Replacement of the wild-type locus was verified by PCR using primers 1072/1216 and 1215/1073 for 5′ and 3′ site-specific integrations, respectively, and full cassette integration was confirmed using primers 1409/1073, which amplified 2.48 kb in TRAP/FlpL, whereas 4.8 kb in *PKAc* cKO. NTC, no template control. **(D)** Parasitemia of *PKAc* cKO in comparison with TRAP/FlpL in mice. Two groups of mice were injected intravenously with either TRAP/FlpL or *PKAc* cKO, and the blood stage growth was monitored by microscopic examination of Giemsa-stained blood smears. Data are shown as means ± SEM (n = 3). One-way ANOVA was used for statistical analysis (P = 0.997).

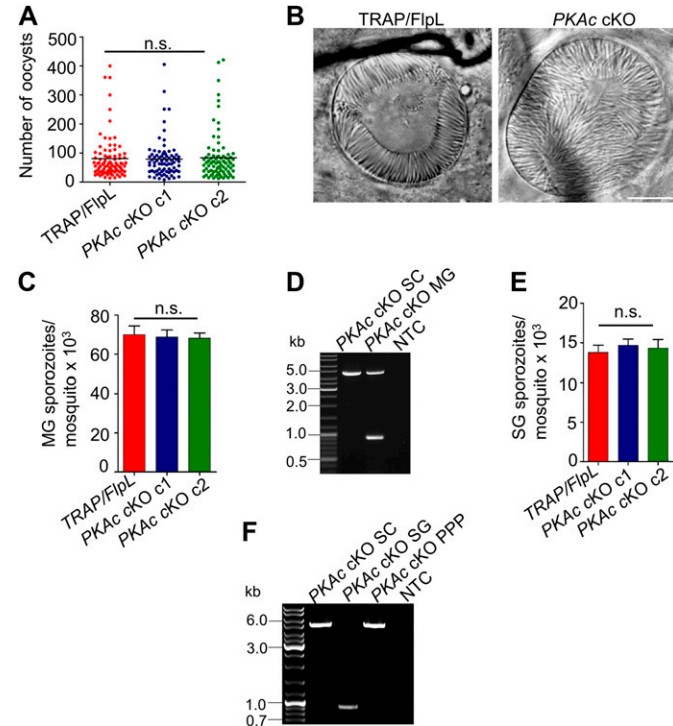

**Figure 2. *PKAc* is not required for parasite development in the mosquito.**
**(A)** Oocysts count per mosquito. One-way ANOVA was used for statistical analysis, and no difference in oocyst numbers was observed (P = 0.923). **(B)** Magnified oocyst showing sporulation. Scale bar, 25 μm. **(C)** Quantifications of the midgut (MG)-associated sporozoites. Data are shown as means ± SEM (n = 3). **(D)** PCR-based confirmation of flirted locus excision in *PKAc* cKO parasites using primers 1409-1073. Both excised and non-excised loci were amplified in *PKAc* cKO MG parasites, but only the intact *PKAc* locus was amplified from the blood stage (SC). **(E)** Quantifications of the SG-associated sporozoites. Data are shown as means ± SEM (n = 3). One-way ANOVA was used for statistical analysis for MG (P = 0.945) and SG (P = 0.806). Error bars represent SEM. **(F)** Flirted *PKAc* cKO locus excision was also monitored in SG and after infecting mice with sporozoites as described previously. The excised locus was amplified in *PKAc* cKO SG parasites, but only the intact *PKAc* locus was amplified from the blood stage (SC) parasites both before (SC) and after passing through mosquito (PPP). MG, midgut; NTC, no template control; PPP, prepatent period; SC, single clone.

monitored by Giemsa-stained blood smears. Control TRAP/FlpL mice became patent on day 3 postinjection (Table 1). However, *PKAc* cKO-injected mice became patent on day 6 with 3 d delay in the prepatent period. The experiment was repeated thrice with the same result. The genotype of the *PKAc* cKO parasites was determined by PCR, which differentiated the excised versus non-excised locus on the basis of size. Genotyping revealed moderate excision of the flirted locus in midgut sporozoites (Fig 2D) and efficient excision in SG sporozoites, but parasites that were carrying the non-excised *PKAc* locus were able to initiate blood stage infection (Fig 2F).

## *PKAc* cKO sporozoites exhibit normal exocytosis and gliding motility and efficiently invade hepatocytes

To understand the stage-specific function(s) of *PKAc*, we systematically investigated the properties of *PKAc* cKO sporozoites. It has been demonstrated that pretreatment of sporozoites with a PKA

**Table 1.   Infectivity of *PKAc* cKO sporozoites in C57BL/6 mice**

| Experiment | Parasite | Number of sporozoites injected | Mice positive/mice injected | Prepatent period (day) |
|---|---|---|---|---|
| 1 | TRAP/FlpL | 5,000 | 4/4 | 3 |
| | *PKAc* cKO c1 | 5,000 | 5/5 | 6 |
| 2 | TRAP/FlpL | 5,000 | 2/2 | 3 |
| | *PKAc* cKO c1 | 5,000 | 3/4 | 6 |
| 3 | TRAP/FlpL | 5,000 | 2/2 | 3 |
| | *PKAc* cKO c2 | 5,000 | 4/4 | 6 |

Mice were inoculated i.v. with *PKAc* cKO or TRAP/FlpL sporozoites. Blood smears were examined daily day 3 p.i., and mice were considered negative if parasites were not detected by day 30.

inhibitor H89 inhibits sporozoite exocytosis and infectivity (Ono et al, 2008). To check exocytosis in *PKAc* cKO sporozoites, the level of circumsporozoite protein (CSP) was determined by immunoblotting. CSP is a very important surface protein implicated in the infectivity of sporozoites. SG sporozoites were incubated in the incubation medium at 37°C and immunoblotting revealed equivalent amounts of secreted CSP in the supernatant in *PKAc* cKO and TRAP/FlpL, suggesting *PKAc* deletion had no effect on exocytosis of sporozoites (Fig 3A). To ensure that identical inputs were present in both groups, pellet fraction was probed with Hsp70 antibody as a loading control (Fig 3A). We next examined whether sporozoites' gliding locomotion was affected by the lack of *PKAc*. For this, *PKAc* cKO SG sporozoites were added to a glass Lab-Tek slide precoated with CSP antibody and trails were visualized using Biotin-labelled CSP antibody followed by streptavidin-FITC (Stewart & Vanderberg, 1988). Counting reveals normal trails in *PKAc* cKO sporozoites as compared with TRAP/FlpL, indicating that PKAc does not have a role in sporozoite gliding motility (Fig 3B and C). To determine invasion of sporozoites, dual color hepatocyte invasion assay was performed which differentiated sporozoites present inside the cell with those present outside the cell (Renia et al, 1988). Sporozoite numbers present outside (red) and total (green) were enumerated (Fig 3D and E), and no significant difference was noted between *PKAc* cKO and TRAP/FlpL sporozoites. These results suggest that PKAc is not required for sporozoite invasion of the host cell.

### *PKAc* cKO parasites develop and egress normally in the liver

To explore, whether PKAc signaling is required for preerythrocytic stages, we systematically investigated the exo-erythrocytic form (EEF) development, parasitophorous vacuolar membrane (PVM) rupture, and merosome formation in *PKAc* cKO parasites in vivo and in vitro. In vivo LS development was quantified at different time points (36, 55, and 72 h) by measuring the parasite 18S rRNA copy number by real-time PCR. Progression in parasite load after infection and decrease after egress from the liver was comparable in *PKAc* cKO and TRAP/FlpL parasites (Fig 4A). To visually observe the development and egress, *PKAc* cKO and TRAP/FlpL sporozoites were allowed to invade hepatocytes and cultures were harvested at 36 and 62 h postinfection. The culture harvested at 36 h postinfection was stained with anti-UIS4 and anti-Hsp70 antibodies, and EEF numbers were enumerated and no difference was observed in the EEF development and numbers (Fig 4B and C). Culture harvested

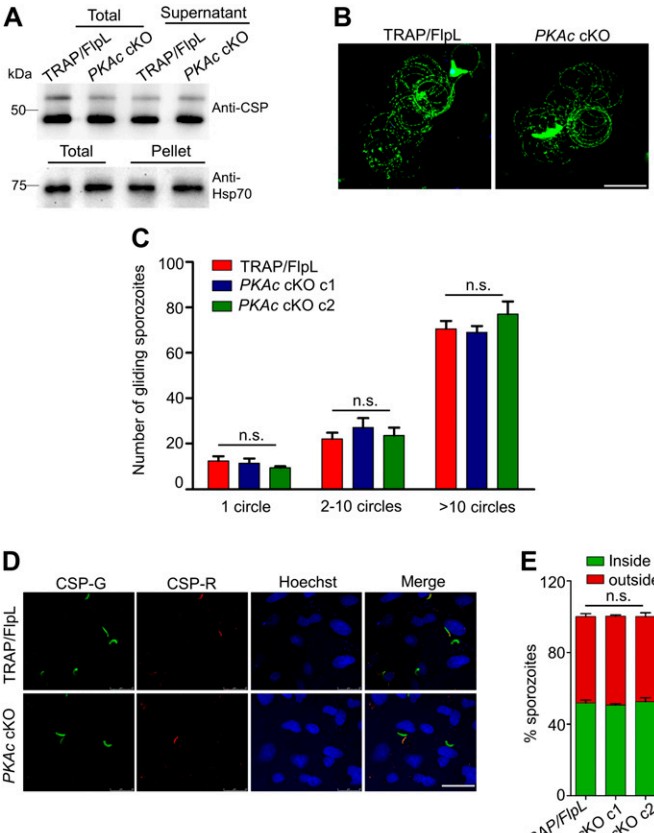

**Figure 3.   Analysis of *PKAc* cKO sporozoites.**
**(A)** *PKAc* cKO sporozoites show normal CSP shedding and secretion in the medium. Hsp70 was used as the loading control. No difference was observed in three independent experiments. **(B, C)** Gliding motility of *PKAc* cKO sporozoites was similar to the gliding activity observed for TRAP/FlpL parasites. Scale bar, 25 μm. Representative data of three independent experiments. One-way ANOVA was used for statistical analysis for 1 (*P* = 0.358), 2–10 (*P* = 0.457), and >10 circles (*P* = 0.273). **(D)** Immunofluorescence assay showing inside/outside sporozoites by differential staining with CSP antibody before and after permeabilization in HepG2 cells. Outside sporozoites were stained before permeabilization with CSP antibody (CSP-R), and total sporozoites were stained after permeabilization (CSP-G). Scale bar, 25 μm. **(E)** Bar graph showing percent sporozoites inside versus outside. Data are shown as means ± SEM (n = 3) in which the total number of sporozoites counted were 2,332 (TRAP/FlpL), 3,527 (*PKAc* cKO c1) and 3,147 (*PKAc* cKO c2). One-way ANOVA was used for statistical analysis (*P* = 0.717).

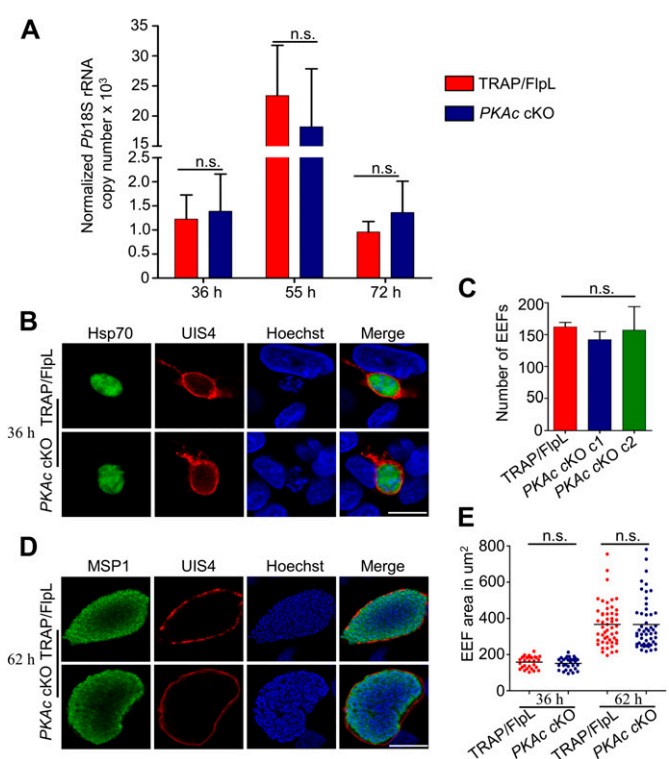

**Figure 4. *PKAc* cKO parasites develop and egress normally in the liver.**
**(A)** To determine if *PKAc* cKO parasites develop and egress from the liver, infected mice livers were harvested at the indicated time points. The parasite burden in the liver was checked by amplifying the parasite 18S rRNA. Mice were treated with a combination of chloroquine, beginning 24 h after sporozoite injection, and artesunate, beginning 39 h after sporozoite injection, to prevent the growth of blood stage parasites. Therefore, the decrease of copy number between 55 and 72 h indicates a loss of liver-specific signal. This experiment was performed twice with similar results, and data from one experiment are presented. Unpaired two-tailed $t$ test was used for statistical analysis for 36 h ($P$ = 0.824), 55 h ($P$ = 0.623), and 72 h ($P$ = 0.493). **(B)** Development of *PKAc* cKO and TRAP/FlpL EEFs in vitro. HepG2 cells were infected with SG sporozoites, and the culture was harvested at different time points. Cultures removed at 36 h p.i. were fixed and parasites were detected using anti-Hsp70 and anti-UIS4 antibodies. Scale bar, 5 $\mu$m. **(C)** Quantification of EEF numbers at 36 h p.i.; no significant difference was observed in three independent experiments. One-way ANOVA was used for statistical analysis ($P$ = 0.693). **(D)** Cultures removed at 62 h p.i. were probed with UIS4 and MSP1 to visualize PV membrane and formation of merozoites, respectively. Scale bar, 5 $\mu$m. **(E)** Quantitation of EEF area at 36 and 62 h p.i. The EEF size was similar in both *PKAc* cKO and TRAP/FlpL parasites. Unpaired two-tailed $t$ test was used for statistical analysis for 36 ($P$ = 0.406), and 62 h ($P$ = 0.977). Error bars represent SD. To determine the size of the EEFs, the perimeter was delineated using the "region of interest" tool and the area was calculated using Nikon NIS elements BR imaging software.

at the late LS time point of 62 h was stained with anti-UIS4 and anti-MSP1 antibodies to monitor the formation of merozoites (Fig 4D). The areas of 36- and 62-h EEFs were measured, and *PKAc* cKO parasites showed robust development with no difference in the size as compared with TRAP/FlpL parasites (Fig 4E). Merozoites are released in the blood after PV membrane rupture in the form of merosomes (Sturm et al, 2006). We quantified the number of merosomes released in the supernatant 65 h p.i. and found that *PKAc* cKO and TRAP/FlpL parasites released comparable numbers of merosomes (Fig 5A). To evaluate the rupture of the PV membrane and merosome morphology in *PKAc* cKO, collected merosomes were also

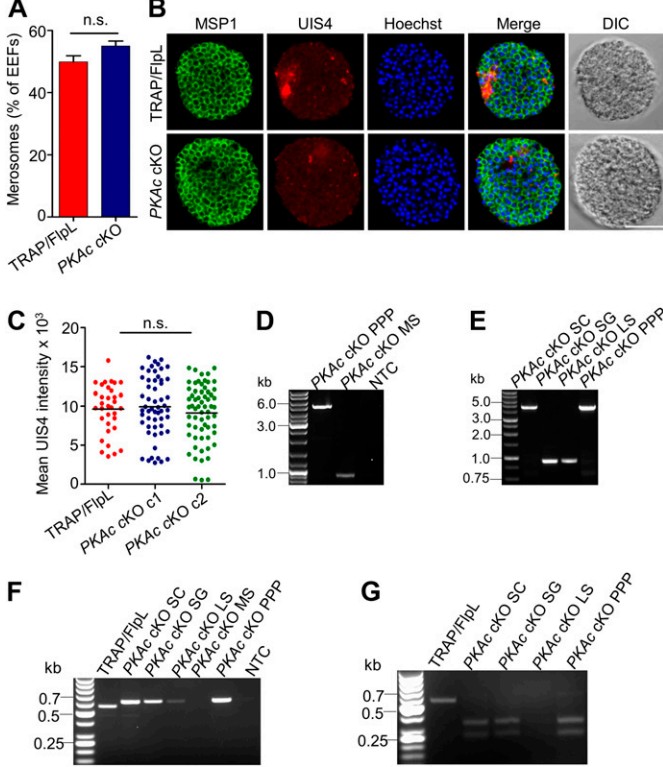

**Figure 5. Merosome development and excision of flirted *PKAc* locus.**
**(A)** *PKAc* cKO parasites released a similar number of merosomes in the culture supernatant as compared with TRAP/FlpL. Merosome experiments were performed three times with similar results, and data from one experiment are presented. Error bars represent SD. Unpaired two-tailed $t$ test was used for statistical analysis ($P$ = 0.104). **(B)** *PKAc* cKO merosomes show normal loss of the PV membrane and normal segregation of merozoite membranes. Scale bar, 5 $\mu$m. **(C)** Measurement of UIS4 intensity showing normal loss of PV membrane in *PKAc* cKO parasites. One-way ANOVA was used for statistical analysis ($P$ = 0.481). **(D, E)** PCR-based confirmation of flirted locus excision in *PKAc* cKO parasites using primers 1409-1073. Genomic DNA was isolated from the TRAP/FlpL line and different stages of *PKAc* cKO parasites. The wild-type locus was amplified from the TRAP/FlpL line, and the excised locus was amplified in *PKAc* cKO SG, LS, and MS parasites, but only the intact *PKAc* locus was amplified from the blood stage (SC) parasites both before (SC) and after passing through mosquito (PPP). **(F)** Amplification of the non-excised region using internal primers 1409-1074. **(G)** Presence of FRT site in the amplified product was confirmed by *Xba*I digestion, which yielded 348- and 255-bp products.

stained with anti-UIS4 and anti-MSP1 antibodies. UIS4 and MSP1 staining showed normal loss of the PV membrane and normal segregation of merozoite membranes, respectively (Fig 5B and C), suggesting that *PKAc* cKO merosomes have normal morphology. Together, these data demonstrate that *PKAc* cKO parasites develop into EEFs and form morphologically normal merosomes.

## PKAc is essential for hepatic merozoite infectivity

To test the infectivity of *PKAc* cKO merozoites, an equal number of TRAP/FlpL and *PKAc* cKO merosomes were injected intravenously into Swiss mice and the development of blood stage parasitemia was monitored. Mice inoculated with different doses of TRAP/FlpL merosomes became positive for blood stage parasites, whereas *PKAc* cKO merosomes failed to initiate blood stage infection (Table 2).

**Table 2. *PKAc* cKO merosomes have impaired infectivity in mice**

| Experiment | Parasite | Number of merosomes injected | Mice positive/mice injected | Prepatent period (d) |
|---|---|---|---|---|
| 1 | TRAP/FlpL | 1 | 5/5 | 5.6 |
| | *PKAc* cKO c1 | 1 | 0/5 | NA |
| | TRAP/FlpL | 10 | 5/5 | 2.2 |
| | *PKAc* cKO c1 | 10 | 0/5 | NA |
| 2 | TRAP/FlpL | 10 | 5/5 | 2 |
| | *PKAc* cKO c1 | 10 | 0/5 | NA |
| | TRAP/FlpL | 10 mr | 5/5 | 2.4 |
| | *PKAc* cKO c1 | 10 mr | 0/5 | NA |
| | TRAP/FlpL | 100 | 5/5 | 1.4 |
| | *PKAc* cKO c1 | 100 | 5/5 | 4.4 |
| | TRAP/FlpL | 100 mr | 5/5 | 1.6 |
| | *PKAc* cKO c1 | 100 mr | 5/5 | 5.6 |
| 3 | TRAP/FlpL | 100 | 5/5 | 1.2 |
| | *PKAc* cKO c1 | 100 | 5/5 | 5.2 |
| | TRAP/FlpL | 100 mr | 5/5 | 1.4 |
| | *PKAc* cKO c1 | 100 mr | 5/5 | 6.2 |

Swiss mice were inoculated i.v. with the indicated number of merosomes with or without mechanical rupture (mr). Blood smears were examined daily as described previously.

Although mice inoculated with high doses of *PKAc* cKO merosomes became positive, they were carrying intact *PKAc* locus as revealed by genotyping (Fig 5D). To investigate the role of PKAc signaling in merosome membrane rupture, we infected mice with mechanically ruptured merosomes by passing through the needle. Mechanical rupture made no significant difference in infection dynamics as compared with non-ruptured merosome infectivity. This implies that PKAc signaling does not play any role in merosome membrane rupture (Table 2). We then checked the genotypes of different stages of *PKAc* cKO parasite. The excised locus was detected in SG, LS, and merosome stage (MS) parasites (Fig 5D and E). Mice that became positive after sporozoite injection showed the presence of only non-excised populations (Fig 5E). The absence of non-excised locus in SG, LS, and MS could be attributed to competitive PCR. The primer set 1409-1074 was designed to amplify a small fragment within the non-excised region, which amplified the non-excised locus from SG, LS, and prepatent (PPP) parasites (Fig 5F). The small fragment was not amplified from MS and that could be due to the low amount of the non-excised parasite. To confirm the presence of the FRT site, the non-excised amplified product was digested with *Xba*I, which is present within the FRT site. Digestion confirmed the presence of the FRT site in different parasite stages (Fig 5G). This indicated that *PKAc* cKO sporozoites contained a mixture of excised and non-excised parasites for the *PKAc* locus. These results demonstrate that PKAc is not important for any stage of preerythrocytic stages, but is essential for erythrocytic stages.

## Discussion

*Plasmodium* parasites sense and respond to the changing host environment for successful infection. Host factors allow parasites to switch from one mode to another like exocytosis and motility is key during invasion, and then it changes to replication mode after switching off both. After successful invasion, the parasite switches to the development phase. Important host factors implicated in parasite activation are low extracellular K⁺ (Dawn et al, 2014), which stimulates microneme secretion in blood stages and high intracellular K⁺ that activates sporozoites for invasion (Kumar et al, 2007; Ono et al, 2008). Despite the identification of these environmental cues, the stage-specific activation of different signaling pathways remained elusive. The essentiality of PKAc in blood stages correlates with its high level of expression in both hepatic and blood merozoites. We also found significant *PKAc* transcripts in SG sporozoites but not as high as in schizonts. To see if sporozoites also rely on cAMP/PKAc signaling, we knocked out *PKAc* using conditional mutagenesis system. Conditional mutant studies provide strong evidence that PKAc activity is required only for blood stages, and its function appears to be redundant in the preerythrocytic stages. We found that deleting *PKAc* in sporozoites did not affect exocytosis and infectivity of sporozoites and egress of merozoites from the merosome, but *PKAc*-deficient merozoites failed to infect erythrocytes. This observation is in contrary to an earlier report in which inhibition of PKAc by H89 affected exocytosis and infectivity of sporozoites (Ono et al, 2008). However, this inhibition could be due to off-target effect, as H89 is known to target multiple kinases, including ribosomal protein S6 kinase β-1, ribosomal protein S6 kinase α-5, rho-associated protein kinase, protein kinase B, and ribosomal protein S6 kinase α-1 (Davies et al, 2000). Other than these, choline kinase is also inhibited by H89 (Wieprecht et al, 1994). In addition, H89 is known to inhibit protein transport from ER to the cell surface (Muniz et al, 1996, 1997). The inability of *PKAc* cKO merozoites to infect erythrocytes is in agreement with indispensability of *PKAc* in the *P. berghei* blood stage (Bushell et al,

2017) and the role of PKAc in merozoite invasion of erythrocytes (Dawn et al, 2014).

PKAc is known to regulate micronemal secretion in *P. falciparum* merozoites (Dawn et al, 2014) and exocytosis in *P. berghei* and *Plasmodium yoelii* sporozoites (Ono et al, 2008). But our results demonstrate that in the absence of *PKAc*, sporozoites secrete CSP, glide normally, and invade hepatocytes efficiently. Furthermore, normal in vitro and in vivo LS developments of *PKAc* cKO parasites suggest that the parasite uses an alternate signaling pathway to activate exocytosis and invasion of the host cell. The role of another kinase PKG has been shown in the regulation of micronemal secretion in *P. falciparum* merozoites (Collins et al, 2013a). Although deletion of PKG had no effect on hepatocyte invasion by the sporozoite, however, parasites failed to mature into merozoites (Falae et al, 2010). Blocking PKG activity by a selective inhibitor, a trisubstituted pyrrole, inhibited both invasion and the egress process of the parasites. The discrepancy between PKG cKO and chemical inhibition was explained by the retention of PKG protein, which was formed before gene excision and was sufficient to initiate signaling during the invasion of hepatocytes. Sporozoites' protein is not retained after transformation of parasites into the LS, which led to the failure of PKG cKO parasites to mature and egress from hepatocytes (Govindasamy et al, 2016). We hypothesize that in *PKAc* cKO parasites, an alternative signaling pathway possibly activates micronemal secretion.

The STRING interaction diagram of *P. berghei* shows the interaction of PKAc with PKG and also with other proteins (Fig S1, Supplemental Data 1, and Table S1). How PKAc and PKG signaling are connected and whether they act through common CDPKs is still unclear. CDPK1 was proposed to play an important role as its disruption was not possible in *P. falciparum* (Kato et al, 2008; Solyakov et al, 2011). It is involved in calcium-triggered microneme secretion of *T. gondii* (Lourido et al, 2010), but its role remained elusive as it is dispensable throughout the life cycle stages of *P. berghei* (Jebiwott et al, 2013). A recent report indicates that CDPK1 regulates the invasion of host RBCs by *P. falciparum* (Kumar et al, 2017). It is also suggested that CDPK1 activity can be turned off by PKAc, although its acting site is not known (Solyakov et al, 2011). Another study suggested that PKG may phosphorylate CDPK1 at S64, and phosphorylation of this site was proposed to target CDPK1 to apical structures (Alam et al, 2015). But CDPK1 S64 can be autophosphorylated, and mutation of this site does not cause a major change in its activity (Ahmed et al, 2012). Therefore, the mechanism of CDPK1 regulation by PKAc and PKG may be distinct and requires further investigation. It was suggested that PKG might compensate for the loss of CDPK1 activity (Bansal et al, 2016). Another kinase CDPK4 deletion results in defective invasion of hepatocytes (Govindasamy et al, 2016), and dispensability of CDPK1 in *P. berghei* suggests that PKAc and PKG possibly function through CDPK4 signaling in the absence of CDPK1. A further detailed investigation is required to understand the crosstalk between PKAc, PKG, and CDPKs during apical regulated exocytosis of parasites.

The other key roles that we checked for PKAc are formation and release of hepatic merozoites from merosomes and invasion of the erythrocyte by hepatic merozoites. In vitro studies confirmed the complete normal development of *PKAc* cKO parasites, which

were similar in number, size, and morphology to control. Fully developed *PKAc* cKO parasites rupture the PV membrane and released merosomes in the culture supernatant, which were comparable to control. Furthermore, *PKAc* cKO parasites invade, develop, and egress from the liver efficiently in vivo. We found that *PKAc* cKO merosomes failed to establish blood stage infection. The high level of *PKAc* transcripts in merozoites concurs their reported role in merozoite invasion. PKAc is highly up-regulated in late LS development before the rupture of the PV membrane. Maturation of EEFs into merozoites and egress from the hepatocyte is an active process tightly controlled by kinases involving a coordinated cascade of events resulting in the activation of proteases (Blackman & Carruthers, 2013; Tawk et al, 2013; Burda et al, 2015). The molecular mechanism of merozoite egress from hepatocytes is currently unclear. *P. berghei* PKG has been implicated in egress of hepatic merozoites (Falae et al, 2010). PKG has been shown to be involved in the secretion of active *P. falciparum* SUB1 from merozoite exonemes into the PVM (Agarwal et al, 2013; Collins et al, 2013b). In *T. gondii*, crosstalk between cAMP and cGMP signaling is reported. Genetic inhibition of PKAc induces premature egress, which is blocked by specific chemical inhibition of PKG (Jia et al, 2017). So it was tempting to check if PKAc activates egress signaling in *P. berghei*. In preerythrocytic stages of *P. berghei*, we show that PKAc is not only dispensable for sporozoite invasion of hepatocytes but also not required for merozoite egress from hepatocytes.

Our work sheds light on the signaling pathways that function during the pre-erythrocytic and erythrocytic stages. The presence of the parasite population mainly carrying excised *PKAc* locus in sporozoites and hepatic stage suggest that PKAc is not required during that stage. Parasites carrying excised *PKAc* locus failed to initiate blood stage infection and parasites which escaped excision were able to establish blood stage infection, confirming the essentiality of PKAc in blood stages. This conclusion is in agreement with recent evidence demonstrating the essential role of PKAc in RBC invasion by *P. falciparum* merozoites (Patel et al, 2019). The absence of non-excised population in the merosomes can be explained by the presence of few merosomes with intact *PKAc* locus, which was not sufficient to detect a non-excised band in PCR although they were present. We demonstrate for the first time the essentiality of *PKAc* in *Plasmodium* blood stages, and more investigation is needed to better understand the role of different kinases and ions during signaling for invasion and egress mechanisms of the parasite. These findings have implications for therapies aimed at preventing erythrocyte invasion by malaria parasites.

# Materials and Methods

### Ethics statement

All animal experiments performed in this study were approved by the Institutional Animal Ethics Committee at Council of Scientific and Industrial Research (CSIR)-Central Drug Research Institute, India (approval no: IAEC/2013/83 and IAEC/2018/3).

 **Life Science Alliance**

### *PKAc* expression analysis by quantitative real-time PCR

For the absolute quantification of *PKAc* transcripts, gene-specific standards were generated by amplifying a 0.12-kb fragment using primers 1307/1308 (primer sequences are given in Table S2). The amplified blunt end fragment was directly cloned into pBluescript SK (+) plasmid at the *EcoR*V site. *P. berghei* Hsp70 gene-specific standards were also generated, and its transcripts were used as an internal control to normalize *PKAc* transcripts (Moreira et al, 2008). Different parasite life cycle stages were prepared as described previously (Al-Nihmi et al, 2017); RNA was isolated using Trizol reagent (15596-026; Invitrogen) and purified using the RNA isolation kit (NP 84105; Genetix) according to the manufacturer's instructions. cDNA was prepared from 2 μg RNA by reverse transcription in a 20-μl reaction mixture containing 1× PCR buffer, 0.5 mM dNTPs, 5 mM MgCl₂, 20 U RNase inhibitor, 2.5 μM random hexamers, and 50 U reverse transcriptase (Applied Biosystems). Real-time PCR was carried out using SYBR green (Bio-Rad), and the ratio of transcript numbers of *PKAc* and *Hsp70* was used to determine the copy number.

### Generation of *PKAc* conditional knockout parasites

For conditional silencing of *PKAc* (PlasmoDB ID: PBANKA_0835600), its intron was flirted. For this, three fragments F1, F2, and F3 having a size of (1.2 kb), (1.6 kb), and (0.53 kb) were amplified using primer pairs 1066/1067, 1068/1069, and 1070/1071, respectively, (primer sequences are given in Table S2). F2 was directly ligated at the *EcoR*V site in the p3'TRAP-hDHFR-flirte vector. The fragment F2 reverse primer (1069) also contained 12 bp of TRAP 3'UTR (bold and italicized in Table S2) for the continuation of UTR function as described previously (Lacroix et al, 2011). After confirmation of correct cloning of F2, fragments F1 and F3 were cloned sequentially at *Sac*II/*Not*I and *Pst*I/*Kpn*I sites, respectively. The integration fragment was separated from the vector backbone by digesting with *Sac*II/*Kpn*I and transfected into the TRAP/FlpL line ([Combe et al, 2009]; S Mishra, KA Kumar, and P Sinnis, unpublished data) as described previously (Janse et al, 2006). For generating a clonal line, recombinant parasites were diluted and intravenously injected into mice. The emerged *PKAc* cKO clonal lines were confirmed by PCR. The parental TRAP/FlpL line served as control in all the experiments.

### Blood growth analysis

To analyze the asexual blood stage propagation, 200 μl of infected blood having 0.2% parasitemia of *PKAc* cKO or TRAP/FlpL was intravenously injected into a group of four Swiss mice. The parasitemia was monitored daily by Giemsa staining of blood smears.

### Analysis of parasite development in mosquito stages

*Anopheles stephensi* mosquito rearing was carried out at 28°C and 80% relative humidity. Mosquitoes were allowed to feed on *PKAc* cKO- or TRAP/FlpL-infected mice to obtain sporozoites. Infected mosquitoes were kept in an environmental chamber maintained at 19°C and 80% relative humidity. On day 14 post blood meal, mosquitoes were dissected and midguts were isolated. Midguts

were observed under a Nikon Eclipse 200 phase contrast microscope with a Nikon Plan 40×/0.65 objective, and oocyst numbers were enumerated. After observing the sporulation pattern, another batch of midguts was crushed and midgut sporozoites were purified after centrifugation and numbers were determined by counting using a hemocytometer. On day 18 post blood meal, mosquitoes were transferred to 25°C to achieve maximum excision of the flirted locus by the FlpL enzyme. To determine the SG-associated sporozoite numbers, glands were isolated, crushed, purified, and counted as described previously (Al-Nihmi et al, 2017).

### Exocytosis

To determine secretion of sporozoite proteins in the supernatant, sporozoites were incubated in the medium containing 1× DMEM, 2.5% FBS, 50 μg/ml Hypoxanthine, and 25 mM Hepes buffer. Sporozoites were incubated for 35 min at 37°C. After incubation, sporozoites were centrifuged at 12,000*g* for 4 min. The pellet and supernatant were separated and SDS–PAGE sample buffer was added. Samples were electrophoresed on 10% polyacrylamide gels and transferred to nitrocellulose membrane by electroblotting. The membrane was blocked, and the supernatant and pellet fractions were probed with an anti-CSP antibody (Yoshida et al, 1980) and anti-Hsp70 antibody (Tsuji et al, 1994), respectively. Bound antibody signals were detected using HRP-conjugated anti-mouse antibodies (W4021; Promega).

### Sporozoite gliding motility assay

To quantify sporozoite gliding motility, a glass eight-well chamber slide was coated with 10 μg/ml anti-CSP antibody in 1× PBS overnight, and the assay was performed as described previously (Stewart & Vanderberg, 1988). SG sporozoites in 3% BSA/DMEM were added 5,000/well and incubated for 1 h at 37°C. After fixation with 4% PFA, trails were visualized by staining with biotinylated anti-CSP antibody, followed by streptavidin-FITC (19538-050; Invitrogen). Trails associated with sporozoites were counted using a Nikon Eclipse 80i fluorescent microscope using a Nikon Plan Fluor 40×/0.75 objective. Pictures of trails associated with sporozoites were acquired using Leica LAS-X software on a confocal laser scanning microscope (Leica TCS SP-8) with an HC PL APO 40×/0.85 objective.

### Sporozoite invasion assay

To differentiate the sporozoites inside the cell versus outside, invasion assay was performed as described (Renia et al, 1988). For this, human liver carcinoma (HepG2) cells were seeded (60,000 cells/well) in a Lab-Tek chamber slide pre-coated with rat-tail collagen type-1. SG sporozoites (20,000/well) were added and immediately centrifuged at 320*g* for 4 min and incubated for 1 h at 37°C in a CO₂ incubator. After incubation, the medium was removed and fixed with 4% paraformaldehyde at room temperature for 20 min followed by washing with PBS twice. Wells were blocked in 1% BSA/PBS followed by staining with anti-CSP mouse monoclonal antibody for 1 h at room temperature. The wells were washed three times with PBS, and the bound CSP signal was revealed by incubation with Alexa-Flour 594 conjugated with anti-mouse IgG.

After completing staining of extracellular sporozoites, cells were permeabilized with chilled methanol for 20 min at 4°C. Blocking and CSP antibody incubation were performed as described above, and the signal was revealed by incubation with Alexa-Fluor 488–conjugated anti-mouse IgG. Finally, wells were incubated with Hoechst 33342 (H13199; Invitrogen) to stain nuclei and mounted in Prolong Diamond antifade reagent (P36970; Life Technologies). Sporozoites were visualized and counted under a Nikon Eclipse 80i fluorescent microscope using a Nikon Plan Fluor 40×/0.75 objective. Images of invading sporozoites were acquired using Leica LAS-X software on a confocal laser scanning microscope (Leica TCS SP-8) with an HC PL APO 40×/0.85 objective.

### Determination of prepatent period

To determine the prepatent period of *PKAc* cKO parasites, SG-isolated sporozoites were inoculated i.v. in C57BL/6 mice. As a control, TRAP/FlpL sporozoites were injected into another group of mice. The parasite in blood was observed by Giemsa staining of blood smears. To check the parasite genotype circulating in blood after breakthrough infection in patent mice, genomic DNA was isolated and PCR was performed using the primer pair 1409/1073 (primer sequences are given in Table S2).

### Quantification of the LS parasite burden

C57BL/6 mice were inoculated i.v. with 5,000 sporozoites as described above. Whole livers were harvested at 36, 55, and 72 h p.i. and homogenized in 10 ml Trizol (Invitrogen); RNA was extracted from 1 ml homogenate following the manufacturer's instructions. cDNA was prepared using cDNA synthesis reagents (Applied Biosystem) as described above. Infection was quantified by real-time PCR, using primers 1195/1196 that amplify *P. berghei* 18S rRNA (Bruna-Romero et al, 2001). For the determination of copy number, plasmid standard curve was run alongside samples. The values of each transcript were normalized to mouse GAPDH (primer sequences are given in Table S2).

### In vitro EEF development and merosome production

To study the EEF development in vitro, HepG2 cells seeded in 48-well plates were infected with sporozoites (5,000/well) as described above, and the rest of the procedure was performed as described previously (Al-Nihmi et al, 2017). Briefly, the medium was changed every 12 h. Coverslips were removed at different stages of EEF development and fixed with 4% paraformaldehyde. For merosome production, 100,000 HepG2 cells were seeded in a 24-well plate and infected with 40,000 sporozoites. The culture supernatant containing merosomes was collected 65 h p.i. and counted using a hemocytometer as described previously (Hopp et al, 2017). Merosomes were injected i.v. into Swiss mice with or without rupturing by passing through a 31-gauge needle 10 times and the prepatent period was determined. To check the merosome genotype, genomic DNA was isolated and PCR was performed as described above. For IFA, merosomes were put into a 12-well slide (ER202W; Thermo Fisher Scientific) and allowed to dry and fixed in 2% PFA.

### Immunofluorescence assay

Both EEFs and merosomes were permeabilized in chilled methanol for 20 min at 4°C and blocked with 1% BSA/PBS. Blocking and antibody incubations were performed for 1 h at room temperature, and rest of the procedures and antibody dilutions were performed as described previously (Al-Nihmi et al, 2017). The culture harvested at 36 h was stained with anti-Hsp70 (Tsuji et al, 1994) and anti-UIS4 (PV marker) (Mueller et al, 2005), and the 62-h culture was stained with anti-UIS4 and anti-merozoite surface protein 1 (MSP1) (Holder & Freeman, 1981) antibodies. Hsp70 and MSP1 signals were revealed by Alexa-Fluor 488–conjugated anti-mouse IgG (A11029; Invitrogen), and the UIS4 signal was revealed by Alexa-Flour 594–conjugated anti-rabbit IgG (A11012; Invitrogen). Nuclei were stained with Hoechst 33342. Merosomes were stained with anti-UIS4 and anti-MSP1 antibodies as previously described for EEFs, and samples were mounted in Diamond antifade reagent. Images were acquired on FV1000 software on a confocal laser scanning microscope (Olympus BX61WI) with an UPlanSAPO 100×/1.4 oil immersion objective. EEF counting was performed manually on a Nikon Eclipse 80i fluorescent microscope using a Plan Fluor 40×/0.75 objective. EEF area and UIS4 intensity measurements were performed using Nikon NIS elements BR imaging software on a Nikon Eclipse 80i fluorescent microscope with a Nikon Plan Fluor 40×/0.75 objective.

### Statistical analysis

Statistical analyses were performed using GraphPad Prism 5 software. One-way ANOVA and unpaired two-tailed *t* test were used to determine the statistical significance.

# Supplementary Information

# Acknowledgements

We thank Dr. Robert Menard (Institute Pasteur) for the p3'trap-hDHFR-flirte3 vector with pUC backbone and Dr. Kota Arun Kumar (University of Hyderabad) for its modified version with pBC backbone, Dr. Photini Sinnis (Johns Hopkins University) and Leiden University Medical Center for *P. berghei* TRAP/FlpL parasite line. We also thank Dr. Anthony A Holder (The Francis Crick Institute) and Dr. Photini Sinnis for MSP1 and UIS4 antibody respectively. We thank Pratik Narain Srivastava for proofreading the manuscript. HH Choudhary and R Gupta acknowledge University Grant Commission for research fellowships. We acknowledge THUNDER (BSC0102) and MOES (GAP0118) Intravital facility for confocal microscopy. S Mishra acknowledges DBT Ramalingaswami Fellowship grant (GAP0142). This manuscript is CDRI communication no. 9850.

### Author Contributions

HH Choudhary: conceptualization, data curation, formal analysis, investigation, methodology, and writing—original draft.

R Gupta: methodology.

S Mishra: conceptualization, data curation, formal analysis, supervision, funding acquisition, validation, investigation, methodology, project administration, writing—original draft, review, and editing.

## Conflict of Interest Statement

The authors declare that they have no conflict of interest.

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
