## [Reviewer comments · Life Science Alliance]

PKAc is not required for the pre-erythrocytic stages of Plasmodium berghei

Hadi Hasan Choudhary, Roshni Gupta, Satish Mishra

DOI: 10.26508/lsa.201900352

Review timeline:

Submission Date:	19 February 2019
Editorial Decision:	13 March 2019
Revision Received:	16 May 2019
Editorial Decision:	21 May 2019
Accepted:	22 May 2019

Report:

(Note: Letters and reports are not edited. The original formatting of letters and referee reports may not be reflected in this compilation.)

No Peer Review Process File is available with this article, as the authors have chosen not to make the review process public in this case.

Thank you for submitting your manuscript entitled "PKAc is essential for erythrocytic stages but not required for pre-erythrocytic stages of *P. berghei*" to Life Science Alliance. The manuscript was assessed by expert reviewers, whose comments are appended to this letter.

As you will see, the reviewers appreciate your data and they provide constructive input on how to further strengthen your study. We would therefore like to invite you to submit a revised version to us, addressing the individual points of the reviewers. We think such a revision is straightforward, but please do get in touch in case you would like to discuss individual revision points further.

Thank you for this interesting contribution to Life Science Alliance. We are looking forward to receiving your revised manuscript.

Thank you for submitting your revised manuscript entitled "PKAc is not required for the pre-erythrocytic stages of *Plasmodium berghei*". As you will see, two of the original reviewers re-assessed your work and appreciate the introduced changes. We would thus be happy to publish your paper in Life Science Alliance pending final revisions necessary to include the recent paper mentioned by reviewer #1 and to meet our formatting guidelines:

- please add a citation to the recent work mentioned by ref#1
- supplementary figures are displayed in-line in the HTML version of your paper; please incorporate the Suppl method and the Suppl Figure legend in the main manuscript and upload Figure S1 and the suppl tables as individual files. The tables should get uploaded as word docx or excel files.

3rd Editorial Decision

22 May 2019

Thank you for submitting your Research Article entitled "PKAc is not required for the pre-erythrocytic stages of *Plasmodium berghei*". It is a pleasure to let you know that your manuscript is now accepted for publication in Life Science Alliance. Congratulations on this interesting work.

Again, congratulations on a very nice paper. I hope you found the review process to be constructive and are pleased with how the manuscript was handled editorially. We look forward to future exciting submissions from your lab.